# ON STRUCTURED STATE-SPACE DUALITY

## ABSTRACT

Structured $\underline{S}$tate-$\underline{S}$pace $\underline{D}$uality (SSD) [Dao & Gu, ICML 2024] is an equivalence between a simple Structured $\underline{S}$tate-$\underline{S}$pace $\underline{M}$odel (SSM) and a masked attention mechanism. In particular, a state-space model with a scalar-times-identity state matrix is equivalent to a masked self-attention with a 1-semiseparable causal mask. Consequently, the same sequence transformation (model) has two algorithmic realizations: a linear-time $O(T)$ recurrence or as a quadratic-time $O(T^2)$ attention. In this work, we formalize and generalize this duality: (i) we extend SSD from the scalar-identity case to general diagonal SSMs (diagonal state matrices); (ii) we show that these diagonal SSMs match the scalar case's training complexity lower bounds while supporting richer dynamics; (iii) we establish a necessary and sufficient condition under which an SSM is equivalent to 1-semiseparable masked attention; and (iv) we provide a negative result that such duality is impossible to extend to standard softmax attention due to rank explosion. Together, these results strengthen the theoretical bridge between recurrent SSMs and Transformers, and widen the design space for expressive yet efficient models.

## 1 INTRODUCTION

Structured $\underline{S}$tate-$\underline{S}$pace $\underline{D}$uality (SSD) refers to a one-to-one equivalence between a certain linear Structured $\underline{S}$tate-$\underline{S}$pace $\underline{M}$odel (SSM) and a masked self-attention mechanism (Dao & Gu, 2024). In plain terms, it means the same sequence transformation has two algorithmic realizations: either as a recurrent state-space system or as a self-attention (matrix) operation. Dao & Gu (2024) introduce the first example of such duality: a state-space model whose state matrix is a scalar multiple of the identity is equivalent to a causal self-attention with a rank-1 mask matrix.

In this case, we write the sequence (time) index of a matrix as a superscript ($A^t$). Then the SSM update $h_{t+1} = A^t h_t + b_t x_t$ (with $h_1 = 0$ and output $y_t = c_t^\top h_t$ for $t = 1, ..., T$) yields the closed-form solution $y_t = \sum_{s=1}^t c_t^\top A^t \cdots A^{s+1} b_s x_s$. Equivalently, the self-attention viewpoint treats $y$ as an explicit attention matrix $M \in \mathbb{R}^{T \times T}$ acting on $x$, with entries $M_{t,s} = c_t^\top A^t \cdots A^{s+1} b_t$ for $s \leq t$ (and $M_{t,s} = 0$ for $s > t$ due to the causal mask). This attention matrix $M$ is a rank-$N$ matrix with a 1-semiseparable (Definition 3.1) causal mask, where $N$ is the dimension of $A^t$, also known as the state dimension. Thus, the SSM and the masked attention realize the same function $x \mapsto y$: one via a linear-time $O(T)$ recurrence and the other via a quadratic-time $O(T^2)$ matrix multiplication.

Remarkably, this duality bridges two disparate paradigms for sequence modeling — recurrent state-space models and Transformer attention. State-space models update a latent state recurrently, and hence yields linear complexity in sequence length. Attention mechanisms compute pairwise token interactions, and hence yields quadratic complexity. The structured state-space duality unifies these two paradigms by revealing that they implement identical functions.

Nevertheless, while Dao & Gu (2024) conjecture that analogues duality should hold for diagonal state-space models, there exist no formal treatment to the best of our knowledge. We give a concrete diagonal state-space duality, and provide the regarding computation algorithm.

**Contributions.**  In this work, we build on SSD and extend its scope in four key directions:

- **General Diagonal SSMs (Sections 4.1 and 4.2).** We extend SSD beyond the simple (scalar) $\times I_N$ state matrix to general diagonal state matrices. This enlarges the class of SSMs under the duality (from a single exponential decay to $N$ separate diagonal dynamics), thereby supporting richer sequence dynamics.

- **Efficiency at Scale (Section 4.3).** We prove that these more general diagonal SSMs match the training complexity lower bound of the scalar case while offering greater expressiveness. In other words, it is possible to train and execute the richer diagonal SSMs with the same optimal $O(TN)$ time complexity as the scalar SSM, so we get additional modeling power at no extra asymptotic cost.

- **Higher-Rank Equivalence (Appendix A).** We prove the conjecture of Dao & Gu (2024) that the equivalence between semiseparable matrices and matrices with sequential state-space representation holds not only for the rank-1 case, but also for general rank $N$. In other words, an SSM of state dimension $N$ corresponds to an $N$-semiseparable attention matrix.

- **Masked Attention Duality of General SSM (Section 4.4).** While each 1-semiseparable masked attention has an SSM dual, we provide a necessary and sufficient condition for an $N$-semiseparable (Definition 3.3) matrix (corresponding to an SSM of state dimension $N$) to have a 1-semiseparable masked attention dual.

Together, these results strengthen the theoretical bridge between recurrent state-space models and Transformer-style attention, and widen the design space for expressive yet efficient sequence modeling. Through this generalized duality framework, we enable principled exploration of new architectures that enjoy the best of both worlds (recurrent and attentional) in terms of speed, capacity, and strong theoretical guarantees.

**Organization.** We provide related work in Section 2, and background in Section 3. We show our main theory in Section 4, and the limitations of structured state-space duality in Section 5.

**Notations.** We denote the index set $\{1, \ldots, I\}$ by $[I]$. We denote vectors with lower case and matrices with upper case. We write the sequence (time) index of a matrix as a superscript ($A^t$). We write the sequence (time) index of a vector or scalar as a subscript ($a_t$). We use $\odot$ for Hadamard multiplication. We write the input sequence of length $T$ as $[x_1^\top, x_2^\top, \ldots, x_T^\top] \in \mathbb{R}^{d \times T}$ and the output at time $t \in [T]$ as $y_t \in \mathbb{R}^{1 \times d}$.

## 2 RELATED WORK

In this section, we review related work on structured state-space models, efficient Transformers and linear attention mechanisms, and the connections between state-space models and attention.

**Structured State-Space Models (SSMs).** SSMs aim to model long-range dependencies with linear-time computation. S3 first introduced the idea of parameterizing state-space models with structured matrices to enable efficient sequence modeling (Gu et al., 2021a). S4 introduces a state-space layer with stable diagonalization and fast convolution, which enabled long-context training and inference (Gu et al., 2021b). S5 simplifies the design with a single multi-input multi-output SSM and preserved $O(T)$ scaling (Smith et al., 2023). Mamba adds input-dependent gating on the SSM projections and achieved strong accuracy with linear-time sequence modeling (Gu & Dao, 2024). These works establish SSMs as competitive sequence models and motivate analyses that compare their expressive power to attention.

**Efficient Transformers and Linear Attention.** Many works reduce the quadratic cost of self-attention by imposing structure or approximation (Tay et al., 2022). Linformer projects keys and values to low rank and reduced compute and memory while preserving accuracy (Wang et al., 2020). Linear Transformers replace softmax by kernel feature maps and execute attention as a recurrence, which yielded $O(T)$ autoregressive inference (Katharopoulos et al., 2020). Performer uses random features to approximate softmax attention with variance control and linear complexity (Choromanski et al., 2021). Nyströmformer applies Nyström approximation to attention and obtains sub-quadratic cost (Xiong et al., 2021). Sparse patterns such as Longformer, BigBird, and Reformer improve scaling by local windows, global tokens, or LSH-based routing (Beltagy et al., 2020; Zaheer et al., 2020; Kitaev et al., 2020). Retentive networks propose a recurrent retention operator that matches Transformer quality with linear-time execution (Sun et al., 2023). These methods show that attention admits efficient surrogates when the attention matrix has a low-rank, kernel, or sparse structure.

**Connections between SSMs and Attention.** Linear attention admits a recurrent implementation and thus links attention with RNN-style computation (Katharopoulos et al., 2020). Dao & Gu (2024) introduce *Structured State Duality* (SSD), prove the duality between scalar-identity SSMs and masked attention with 1-semiseparable kernels, and conjectrue such duality hold in diagonal SSM. Hydra generalizes matrix mixers beyond causal SSMs with quasi-separable structure and bidirectional information flow (Hwang et al., 2024).

Our work differs in scope and goal: we give an algebraic duality between $N$-dimensional diagonal SSM and 1-semiseparable masked attention, and we prove that diagonal SSMs match the scalar case in training FLOPs and memory while enabling $N$ independent state modes. These results place diagonal SSD on a firm foundation and clarifies how SSM capacity aligns with semiseparable rank.

## 3 BACKGROUND

In this section, we present the foundational concepts and definitions in Section 3.1. Then we provide the existing theory from (Dao & Gu, 2024) for the structured state-space duality in Section 3.2.

### 3.1 SEMISEPARABLE MATRIX DEFINITIONS

We begin by defining the class of semiseparable (SS) matrices that prepares our theoretical development. First, we recall the base case of 1-semiseparable matrices:

**Definition 3.1** (1-Semiseparable (1-SS) Matrix.). *Suppose $M$ is a lower triangular matrix. $M$ is 1-semiseparable (1-SS) if and only if every submatrix of $M$ consisting of entries on or below the main diagonal has rank at most 1.*

Intuitively, a 1-SS matrix has extremely low complexity: each new row introduces at most one new independent direction in the space of lower-triangular entries. We next define a masked attention operator that is structured by such a matrix.

**Definition 3.2** (1-SS Masked Attention.). *Let $Q, K \in \mathbb{R}^{T \times N}$ be query and key matrices. A 1-SS masked attention is a self-attention operation whose attention weight matrix is masked by a 1-SS matrix $M \in \mathbb{R}^{T \times T}$. In particular, the attention scores take the form $M \odot (QK^\top)$, i.e. the element-wise product of $QK^\top$ with the mask $M$ (with $M$ enforcing a causal lower-triangular structure).*

We now generalize from 1-semiseparable to (higher-rank) $N$-semiseparable as follows. In essence, an $N$-semiseparable matrix allows up to $N$ independent directions in each lower-triangular block.

**Definition 3.3** ($N$-Semiseparable ($N$-SS) Matrix.). *A lower triangular matrix $M$ is $N$-semiseparable (N-SS) if every submatrix of $M$ consisting of entries on or below the main diagonal has rank at most $N$. The smallest such $N$ is called the semiseparable* rank *(or order) of $M$.*

Furthermore, we introduce the notion of a *Sequentially SemiSeparable* (SSS) representation. Importantly, SSS connects these structured matrices to state-space models:

**Definition 3.4** ($N$-Sequentially Semiseparable ($N$-SSS) Representation.). *A lower triangular matrix $M \in \mathbb{R}^{T \times T}$ has an $N$-sequentially semiseparable (N-SSS) representation if there exist vectors $b_1, \ldots, b_T \in \mathbb{R}^N$, $c_1, \ldots, c_T \in \mathbb{R}^N$, and matrices $A^1, \ldots, A^T \in \mathbb{R}^{N \times N}$ such that*

$$M_{j,i} = c_j^\top A^j \cdots A^{i+1} b_i, \tag{3.1}$$

*for all $1 \leq i \leq j \leq T$.*

**Definition 3.5** ($N$-SSS Representable Matrix.). *A matrix $M$ is $N$-SSS representable if it admits an $N$-SSS representation (Equation (3.1)). Equivalently, $M$ can be written in the form of (3.1).*

The above definitions formalize how a structured state-space model of dimension $N$ gives rise to a matrix $M$ with semiseparable rank $N$. In particular, any $M$ that has an $N$-SSS representation is necessarily $N$-semiseparable (since each new state dimension contributes at most one new rank to the growing matrix). We next review the known correspondence between such structured matrices and attention mechanisms in the simplest (rank-1) case.

## 3.2 EXISTING STRUCTURED STATE-SPACE DUALITY

We now describe the structured state-space duality as originally established by Dao & Gu (2024) for the scalar-identity state-space case. We begin by formulating the state-space model and its induced sequence kernel, then show how it corresponds to a masked attention operator.

**Time-Varying SSM and Induced Kernel.** Consider a time-varying linear state-space model (SSM) with state dimension $N$, defined by the recurrence

$$\underbrace{h_t}_{N \times d} := \underbrace{A^t}_{N \times N} \underbrace{h_{t-1}}_{N \times d} + \underbrace{b_t}_{N \times 1} \underbrace{x_t}_{1 \times d}, \quad \underbrace{y_t}_{1 \times d} = \underbrace{c_t^\top}_{1 \times N} \underbrace{h_t}_{N \times d}, \quad \text{for} \quad t \in [T], \tag{3.2}$$

where we set $h_0 = 0$ for consistency. Here, $h_t \in \mathbb{R}^{N \times d}$ is the hidden state, $A^t \in \mathbb{R}^{N \times N}$ is the state transition matrix, $b_t \in \mathbb{R}^{N \times 1}$ and $c_t \in \mathbb{R}^{N \times 1}$ are input and output weight matrices. Importantly, this recurrence defines a causal linear operator on the input sequence. Unrolling the recurrence yields an explicit input-output relation

$$y_t = \sum_{s=1}^{t} M_{t,s} x_s, \quad \text{where} \quad M_{t,s} := \begin{cases} c_t^\top A^t \cdots A^{s+1} b_s, & \text{for} \quad t \geq s; \\ 0, & \text{for} \quad t < s. \end{cases} \tag{3.3}$$

for $1 \leq s \leq t \leq T$. We refer to $M_{t,s}$ as the *SSM kernel* at $(t, s)$. Let $M \in \mathbb{R}^{T \times T}$ denote the lower-triangular matrix of kernel coefficients, i.e. $M_{t,s}$ for $t \geq s$ (and $M_{t,s} = 0$ for $t < s$). By construction, $M$ encodes the entire transformation from inputs $x_1, \ldots, x_T$ to outputs $y_1, \ldots, y_T$. Moreover, $M$ is structured: since the latent state is $N$-dimensional, $M$ has semiseparable rank at most $N$ (each row of $M$ lies in an $N$-dimensional subspace). In particular, $M$ is an $N$-SS matrix in the sense of Definition 3.3, and for this special case $N = 1$, $M$ is 1-SS.

**Scalar-Times-Identity State Matrix** ($A^t = a_t I_N$). A particularly simple case of the above is when each state matrix is a scalar multiple of the identity. We call such an SSM a *scalar-identity SSM*, meaning $A_t = a_t I_N$ for some scalar $a_t \in \mathbb{R}$. In this case, the recurrence (3.2) simplifies to

$$y_t = \sum_{s=1}^{t} a_t \cdots a_{s+1} c_t^\top b_s x_s, \tag{3.4}$$

which is a convolution-style sum over past inputs. For example, if $a_t = a$ is constant, then (3.4) reduces to the standard discrete-time convolution $y_t = \sum_{s=1}^{t} a^{t-s} c_t^\top b_s x_s$.

**Scalar-Identity SSM.** We call an SSM layer a scalar-identity SSM if each of the state matrices $A^t$ is a scale multiple of the identity matrix.

**Rank-1 Special Case** ($N = 1$). In the extreme case of state dimension $N = 1$, the state $h_t$ is one-dimensional. Then $b_t$ and $c_t$ are scalars for all $t$. We can collect the input sequence into a matrix $X = [x_1; x_2; \ldots; x_T] \in \mathbb{R}^{T \times d}$ (with $x_t^\top$ as the $t$-th row) and similarly $Y = [y_1; \ldots; y_T] \in \mathbb{R}^{T \times d}$ for the outputs. Let $p = (b_1, \ldots, b_T)^\top \in \mathbb{R}^T$ and $q = (c_1, \ldots, c_T)^\top \in \mathbb{R}^T$ denote the vectors of input and output weights over time. The scalar-identity formula (3.4) then reduces to

$$\underbrace{Y}_{T \times d} = \underbrace{\text{diag}(p)}_{T \times T} \underbrace{M}_{T \times T} \underbrace{\text{diag}(q)}_{T \times T} \underbrace{X}_{T \times d},$$

where

$$M_{t,s} := \begin{cases} a_t \cdots a_{s+1}, & \text{for} \quad t \geq s; \\ 0, & \text{for} \quad t < s. \end{cases}$$

Here $M$ is a 1-semiseparable mask matrix (Definition 3.1). In other words, the sequence mapping implemented by this $N = 1$ SSM can be viewed as a masked attention operation: $M$ serves as a causal mask on the outer-product matrix $CB^\top = qp^\top$. In the notation of Definition 3.2, this is a 1-SS masked attention with $Q = C$ and $K = B$.

Now we are ready to present the structured state-space duality by Dao & Gu (2024):

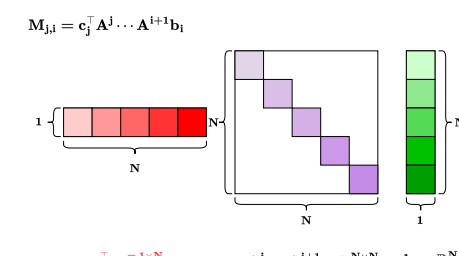

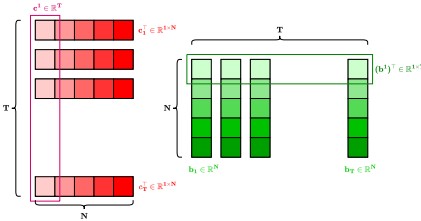

Figure 1: $M_{j,i} = c_j^\top A^j \cdots A^{i+1} b_i$      Figure 2: Construction of $b^n$ and $c^n$.

**Proposition 3.6** (Dao & Gu (2024) Scalar-Identity State-Space Duality.). *Consider the SSM defined by (3.2) where each $A^t = a_t I_N$ (i.e. a scalar-identity SSM). Let $B = [b_1; b_2; \ldots; b_T]^\top \in \mathbb{R}^{T \times N}$ and $C = [c_1; c_2; \ldots; c_T]^\top \in \mathbb{R}^{T \times N}$ be the matrices whose $t$-th rows are $b_t^\top$ and $c_t^\top$, respectively. Define $M \in \mathbb{R}^{T \times T}$ by $M_{t,s} = a_t a_{t-1} \cdots a_{s+1}$ for $t \geq s$ and $M_{t,s} = 0$ for $t < s$. Then for any input sequence $X = [x_1; \ldots; x_T] \in \mathbb{R}^{T \times d}$ with output $Y = [y_1; \ldots; y_T] \in \mathbb{R}^{T \times d}$, the recurrence (3.4) is equivalent to a 1-SS masked attention representation:*

$$Y = (M \odot (CB^\top))X,$$

*where*

$$M_{t,s} := \begin{cases} a_t \cdots a_{s+1}, & \text{for} \quad t \geq s; \\ 0, & \text{for} \quad t < s. \end{cases}$$

*Here $\odot$ denotes elementwise (Hadamard) product. In particular, the same sequence transformation is realizable either by the linear-time recurrence (3.2) or by the quadratic-time matrix operation on the right-hand side.*

Proposition 3.6 (from Dao & Gu (2024)) establishes a one-to-one correspondence between a simple structured SSM and a masked self-attention operator with a 1-SS (rank-1) mask.

## 4 MAIN THEORY

In this section, we provide the structured state-space duality for general diagonal SSMs in Section 4.1, structured state-space duality for diagonal SSMs with full-rank state matrices in Section 4.2, computational complexity of diagonal SSD in Section 4.3, and general SSMs having 1-SS masked attention dual in Section 4.4.

### 4.1 STRUCTURED STATE-SPACE DUALITY FOR GENERAL DIAGONAL SSMS

While Dao & Gu (2024) only study state-space duality of SSM with scalar-identity state matrices, we extend state-space duality to SSM with general diagonal state matrices.

In the case of general diagonal SSM, where each $A^t$ is a diagonal matrix in (3.1), the state-space model also has an attention-like dual.

**Attention-Like Dual of Diagonal SSMs.** Suppose $M \in \mathbb{R}^{T \times T}$ is a lower triangular matrix as in (3.1) regarding the state-space model. We show that $M$ has an attention-like dual as the sum of $N$ attention-like matrices $M^n = L^n \odot (Q^n \cdot K^{n\top})$, where for all $n \in [N]$ we have $Q^n, K^n \in \mathbb{R}^{T \times 1}$.

Specifically, suppose

$$M_{j,i} = c_j^\top A^j \cdots A^{i+1} b_i \tag{4.1}$$

for all $1 \leq i \leq j \leq T$, where $b_1, \cdots, b_T, c_1, \cdots, c_T \in \mathbb{R}^N$ and each $A^t \in \mathbb{R}^{N \times N}$ is a diagonal matrix. See Figure 1.

Then we have $M_{j,i} = \sum_{n=1}^N (c_j)_n (A^j \cdots A^{i+1})_{n,n} (b_i)_n$ for all $1 \leq i \leq j \leq T$.

Note that those terms are separated for different $n_1, n_2 \in [N]$. For all $n \in [N]$, let $b^n, c^n \in \mathbb{R}^T$ be such that for all $t \in [T]$, $b_t^n = (b_t)_n$ and $c_t^n = (c_t)_n$. See Figure 2.

Define $1\mathrm{SS}(\cdot) : \mathbb{R}^T \to \mathbb{R}^{T \times T}$ by

$$1\mathrm{SS}(a_1, a_2, \cdots, a_T) = \underbrace{M}_{T \times T},$$

where

$$M_{t,s} := \begin{cases} a_t \cdots a_{s+1}, & \text{for} \quad t \geq s; \\ 0, & \text{for} \quad t < s. \end{cases}$$

for $1 \leq t, s \leq T$. Then we verify that $M = \sum_{n=1}^N M^n$, where $M^n = 1\mathrm{SS}(A_{n,n}^1, \cdots, A_{n,n}^T) \odot (b^n \cdot c^{n\top})$ with simple algebra.

## 4.2 STRUCTURED STATE-SPACE DUALITY FOR DIAGONAL SSMs WITH FULL-RANK STATE MATRICES

Now we use the attention-like representation of $M$ in Section 4.1 to construct the 1-SS masked attention dual of $M$. When all the state matrices $A^t$ of the state-space model have full rank, the attention-like dual of diagonal SSM turns into 1-SS masked attention dual.

**1-SS Attention Dual of SSM with Full-Rank Diagonal State Matrices.** Suppose $M \in \mathbb{R}^{T \times T}$ has $N$-SSS representation as in (3.1), where each $A^t$ is a diagonal matrix with none-zero determinant. In this case we show that $M$ has a 1-SS masked attention dual.

Specifically, when $\det(A^t) \neq 0$ for all $t \in [T]$, $M^n$ has the representation of

$$1\mathrm{SS}(1, 1, \cdots, 1) \odot (b'^n \cdot c'^{n\top}),$$

where $b_t'^n = b_t^n \cdot (A_{n,n}^1 \cdots A_{n,n}^t)$ and $c_t'^n = c_t^n / (A_{n,n}^1 \cdots A_{n,n}^t)$ for all $t \in [T]$. Let $B', C' \in \mathbb{R}^{T \times N}$ be such that $B'_{:,n} = b'^n$ and $C'_{:,n} = c'^n$ for all $n \in [N]$, then $M = 1\mathrm{SS}(1, 1, \cdots, 1) \odot (B' \cdot C'^{\top})$.

## 4.3 COMPUTATIONAL COMPLEXITY OF DIAGONAL SSD

We give the concrete computation algorithm of diagonal state-space duality and evaluate its efficiency in aspects of computation cost, total memory and parallelization.

**Computation Algorithm.** Define $f : \mathbb{R}^T \times \mathbb{R}^{T \times d} \to \mathbb{R}^{T \times d}$ by $f(x, Y)_{:,s} = x \odot (Y_{:,s})$ for all $s \in [d]$. Define $g : \mathbb{R}^T \times \mathbb{R}^{T \times d} \to \mathbb{R}^{T \times d}$ by $g(x, Y)_{1,:} = Y_{1,:}$, $g(x, Y)_{t+1,:} = x_{t+1} \cdot g(x, Y)_{t,:} + Y_{t+1,:}$ for $t \in [T-1]$. Consider the SSM layer with state dimension $N$ defined by (3.2), where each $A^t$ is a diagonal matrix. This recurrence relation also has representation

$$\underbrace{Y}_{T \times d} = \underbrace{M}_{T \times T} \cdot \underbrace{X}_{T \times d},$$

where

$$M_{j,i} = c_j^\top A^j \cdots A^{i+1} b_i.$$

Express $M$ as $M = \sum_{n=1}^N M^n$, where $M^n = 1\mathrm{SS}(A_{n,n}^1, \cdots, A_{n,n}^T) \odot (b^n \cdot c^{n\top})$. Let $a^n \in \mathbb{R}^T$ denote $(A_{n,n}^1, \cdots, A_{n,n}^T)$ for $n \in [N]$. Denote $Y = M \cdot X$ as $Y = \mathrm{SSM}(X)$. Then $\mathrm{SSM}(X)$ is computed as the following algorithm Algorithm 1.

**Computation Cost.** Since each step of Algorithm 1 takes computation cost of $\mathcal{O}(NTd)$, this algorithm takes total computation cost of $\mathcal{O}(NTd)$ FLOPs.

**Total Memory Cost.** The memory cost of the state data $A^1, \cdots, A^T, b_1, \cdots, b_T, c_1, \cdots, c_T$ is $TN + TN + TN = \mathcal{O}(NT)$. In the first three steps of Algorithm 1, each step generates $n$ matrices of size $T \times d$, and in the lsat step of Algorithm 1, only one matrix of size $T \times d$ is generated. Therefore the memory cost of the intermediate step is $NTd + NTd + NTd + Td = \mathcal{O}(NTd)$. Considering all the memory costs above, we deduce that diagonal state-space duality has total memory cost $\mathcal{O}(NT) + \mathcal{O}(NTd) = \mathcal{O}(NTd)$.

---

**Algorithm 1** Diagonal state space dual(SSD)

> **procedure** SSM($X$)
> $\quad Z^n \leftarrow f(b^n, X) \quad$ for all $n \in [N]$ $\qquad\qquad\qquad\qquad$ ▷ Time $\mathcal{O}(NTd)$
> $\quad H^n \leftarrow g(a^n, Z^n) \quad$ for all $n \in [N]$ $\qquad\qquad\qquad\quad$ ▷ Time $\mathcal{O}(NTd)$
> $\quad Y^n \leftarrow f(c^n, H^n) \quad$ for all $n \in [N]$ $\qquad\qquad\qquad\quad$ ▷ Time $\mathcal{O}(NTd)$
> $\quad Y \leftarrow \sum_{n=1}^{N} Y^n$ $\qquad\qquad\qquad\qquad\qquad\qquad\qquad\quad$ ▷ Time $\mathcal{O}(NTd)$
> $\quad$ **return** $Y$
> **end procedure**

---

**Parallelization.** Note that the first 3 steps of Algorithm 1 are operated in parallel for each $n \in [N]$, the diagonal state-space dual has separation into $N$ parallel computation processes, each of them costing time $\mathcal{O}(Td)$. Furthermore, note from the definition of $f$ and $g$ that the all columns of $X$ are operated respectively during the whole processing of Algorithm 1. Therefore the diagonal state-space dual has further separation into $Nd$ parallel computation processes, each process costing time $\mathcal{O}(T)$.

### 4.4 General SSMs Having 1-SS Masked Attention Dual

We further study the duality between 1-SS masked attention and general SSM.

**Equivalence Between $N$-SS Matrices and $N$-SSS Representable Matrices.** Firstly we state the equivalence between the class of $N$-SS matrices and the class of $N$-SSS representable matrices. Note that there exists a trivial 1-1 correspondence between SSMs and $N$-SSS representations.

**Proposition 4.1** (Proposition 3.3 in (Dao & Gu, 2024)). *A lower triangular matrix is $N$-semiseparable iff it is $N$-SSS representable.*

*Proof.* For detailed proof, see Appendix A. $\qquad\qquad\qquad\qquad\qquad\qquad\qquad\qquad$ □

**Remark 4.2.** *We remark that Proposition 4.1 complements the proof of Dao & Gu (2024, Proposition 3.3). Our constructive proof reveals more details and gives a concrete method to derive the corresponding $N$-SSS representation from an $N$-SS matrix.*

**Remark 4.3.** *Versions of this equivalence appear in the structured-matrix literature (semiseparable/quasiseparable/SSS). We include a self-contained constructive proof tailored to the causal setting (connecting to attention mechanism in transformer architectures). This makes the result accessible to the ML audience and to enable our higher-rank SSD instantiation.*

**SSMs Having 1-SS Masked Attention Dual.** Now that we have the equivalence between $N$-SS matrices and $N$-SSS representable matrices, we use $N$-SS matrices to study the duality between 1-SS masked attention and general SSM. We provide a necessary and sufficient condition for an SSM to have 1-SS masked attention dual regarding to the SSM's corresponding attention matrix.

Suppose $M \in \mathbb{R}^{T \times T}$ is an $N$-SS matrix. We study the necessary and sufficient condition for $M$ to have a 1-SS masked attention dual.

**Definition 4.4** (Fine 1-SS Matrix.). *We say a 1-SS matrix $L = 1SS(a_1, a_2, \cdots, a_t)$ is a fine 1-SS matrix iff $a_1 a_2 \cdots a_t \neq 0$.*

**Definition 4.5** (New Column of Lower Triangular Matrix.). *We call $M_{t:,t}$ a new column of $M$ iff $M_{t:T+1,t}$ is not in $M_{t:,:t}$'s column space.*

**Proposition 4.6.** *Suppose $M \in \mathbb{R}^{T \times T}$ is an $N$-SS lower triangular matrix. Then $M$ has representation of 1-SS masked attention $L \odot (QK^\top)$ for some $Q, K \in \mathbb{R}^{T \times N}$ and fine 1-SS matrix $L$ iff it has at most $N$ new columns.*

*Proof.* The proof consists of two parts.

**Part 1.** In this part we show that $M$ does not have representation of fine 1-SS masked attention if it has more than $N$ new columns.

Suppose $M = L \odot QK^\top$ for some $Q, K \in \mathbb{R}^{T \times N}$ and $L = 1SS(a_1, a_2, \cdots, a_T)$ where $a_1 a_2 \cdots a_T \neq 0$. We then multiply the $t$-th row by $\frac{1}{a_1 a_2 \cdots a_t}$ and multiply the $t$-th column by

$a_1 a_2 \cdots a_t$ for all $t \in [T]$. Note that these operations don't change the number of new columns of the matrix.

After these operations we get a lower triangular matrix $M'$ having at least $N + 1$ new columns. The lower triangular part of $M'$ is exactly the same as the lower triangular part of $QK^\top$. Denote $W := QK^\top$, then $W$ has rank at most $N$.

Suppose $M'$ has new columns $M'_{t_1:,t_1}, M'_{t_2:,t_2}, \cdots M'_{t_{N+1}:,t_{N+1}}$ for $1 \le t_1 < t_2 < \cdots < t_{N+1} \le T$.

We claim that $W_{:,t_1}, W_{:,t_2}, \cdots, W_{:,t_{N+1}}$ are linearly independent. If not so, there exist $c_1, c_2, \cdots, c_{N+1} \in \mathbb{R}$ such that at least one of them is none-zero and $c_1 W_{:,t_1} + c_2 W_{:,t_2} + \cdots + c_{N+1} W_{:,t_{N+1}} = \mathbf{0}_T$.

Suppose $n$ is the largest index in $[N+1]$ such that $c_{t_n} \ne 0$. Then we have $c_1 W_{:,t_1} + c_2 W_{:,t_2} + \cdots + c_n W_{:,t_n} = \mathbf{0}_T$.

This implies that $c_1 M'_{t_n:,t_1} + c_2 M'_{t_n:,t_2} + \cdots + c_n M'_{t_n:,t_n} = c_1 W_{t_n:,t_1} + c_2 W_{t_n:,t_2} + \cdots + c_n W_{t_n:,t_n} = \mathbf{0}_{T-t_n+1}$, which contradicts to the fact that $M'_{t_n:,t_n}$ is a new column of $M'$.

Then we deduce that $W_{:,t_1}, W_{:,t_2}, \cdots, W_{:,t_{N+1}}$ are linearly independent. This implies that $W$ has rank at least $N + 1$, which contradicts to $W = QK^\top$. Therefore $M$ doesn't have the representation of $L \odot (QK^\top)$ where $Q, K \in \mathbb{R}^{T \times N}$ and $L$ is a fine 1-SS matrix.

**Part 2.** In this part we show that any lower triangular matrix with at most $N$ new columns has representation of $L \odot (QK^\top)$ for some $Q, K \in \mathbb{R}^{T \times N}$ and fine 1-SS matrix $L \in \mathbb{R}^{T \times T}$.

Suppose $M \in \mathbb{R}^{T \times T}$ is a lower triangular matrix having at most $N$ new columns. We now change $M$'s entries above the diagonal to create a matrix with rank at most $N$.

We change the entries column by column from the left to the right.

For $t \in [T]$, if $M_{t:,t}$ is a new column of $M$, remain $M'_{:t,t}$ to be $\mathbf{0}_{t-1}$; if $M_{t:,t}$ is not a new column of $M$, there exist $c_1, c_2, \cdots, c_{t-1} \in \mathbb{R}$ satisfying

$$M_{t:,t} = \sum_{s=1}^{t-1} c_s M t:, s.$$

Set $M'_{:t,t}$ to be

$$\sum_{s=1}^{t-1} c_s M'_{:t,s},$$

then $M'_{:,:t+1}$ and $M'_{:,:t}$ have the same column rank.

Given that $M$ has no more than $N$ new columns, we deduce by mathematical induction that $M'$ has rank at most $N$. Therefore there exist $Q, K \in \mathbb{R}^{T \times N}$ such that $M' = QK^\top$.

Then we have $M = 1\text{SS}(1, 1, \cdots, 1) \odot (QK^\top)$.

This completes the proof. $\qquad\square$

The following results are deduced from Proposition 4.6.

**Lemma 4.7.** *Suppose $M \in \mathbb{R}^{T \times T}$ is an $N$-SS lower triangular matrix. Then $M$ has representation of $1$-SS masked attention $L \odot (QK^\top)$ for some $Q, K \in \mathbb{R}^{T \times N}$ iff $M$ has several diagonal blocks containing all the none-zero entries of $M$, and each of the diagonal blocks has at most $N$ new columns.*

**Theorem 4.8.** *Suppose $M$ is an $N$-SS matrix corresponding to an SSM. This SSM has $1$-SS masked attention dual iff $M$ has several diagonal blocks containing all the none-zero entries of $M$, and each of the diagonal blocks has at most $N$ new columns.*

**Remark 4.9.** *In Proposition 4.6 we focus on fine $1$-SS masked attention and in Lemma 4.7 our conclusion holds for general $1$-SS masked attention, where for the causal mask $L = 1SS(a_1, a_2, \cdots, a_T)$, $a_t$ is possibly $0$ for some $t \in [T]$.*

## 5 LIMITATIONS OF STRUCTURED STATE-SPACE DUALITY

We study the limitation of state-space duality from two sides. (i) From the attention side, we show impossibility of extending SSD to softmax attention; (ii) from the state-space model (SSM) side, we show impossibility of extending to general SSM with low state dimension.

**Impossibility of Extending SSD to Softmax Attention.** We provide a trivial example to show that softmax attention does not have state-space duality. Consider a matrix $V \in \mathbb{R}^{T \times T}$ such that $V_{ij} = i \times j$ for all $i, j \in [T]$. The matrix $V$ has rank 1 because each of its column vectors is a multiple of $(1, 2, \cdots, T)^\top \in \mathbb{R}^T$. However, $\mathrm{softmax}(V)$ has rank $T$ according to the Vandermonde determinant, and furthermore, each submatrix of $\mathrm{softmax}(V)$ is full rank. This implies that even when the attention matrix $QK^\top$ has very low rank, the rank of $\mathrm{softmax}(QK^\top)$ expands to $T$ in most cases. Moreover, any attention matrix that has a state-space dual must be $N$-semiseparable, where $N$ is the state dimension of the corresponding state-space model. Therefore, softmax attention does not have a state-space dual.

**Impossibility of Extending SSD to General SSM with Low State Dimension.** We provide an example to show that general SSM doesn't have a state-space dual, even when the state dimension is very low.

**Proposition 5.1.** *Consider the SSM layer with state dimension $N \geq 2$ defined by (3.2), there exist $A^{1:T+1}, b_{1:T+1}, c_{1:T+1}$ such that the recurrence relation doesn't have an attention dual.*

*Proof.* According to Proposition 4.1, there exist $A^{1:T+1}, b_{1:T+1}, c_{1:T+1}$ such that the recurrence relation (3.2) has representation $Y = M \cdot X$, where $M = I_{T \times T} + E^{T,1}$ is a 2-SS matrix. Here

$$E_{j,i}^{T,1} = \begin{cases} 1, & j = T \text{ and } i = 1; \\ 0, & \text{otherwise.} \end{cases}$$

We claim that $M$ doesn't have the representation of $L \odot (QK^\top)$ where $Q, K \in \mathbb{R}^{T \times N}$ and $L$ is a 1-SS matrix.

Otherwise, suppose

$$L_{j,i} = \begin{cases} a_j \cdots a_{i+1}, & \text{for } j \geq i; \\ 0, & \text{for } j < i. \end{cases}$$

for $i, j \in [T]$.

Since $M_{T,1} = 1$, $L_{T,1} = a_2 \cdots a_T$ is none-zero, i.e. each of $a_2, a_3, \cdots, a_T$ is none-zero. From this we deduce that for all $1 \leq j < i \leq T - 1$, $(QK^\top)_{i.j} = 0$.

Since each diagonal element of $M$ is none-zero, each diagonal element of $QK^\top$ is also none-zero. Given that $(QK^\top)_{i.j} = 0$ for all $1 \leq j < i \leq T - 1$, we deduce that $QK^\top$ has rank at least $T - 1$, which is a contradiction. $\square$

## 6 DISCUSSION AND CONCLUSION

We initiate a unified framework that reveals deep structural parallels between recurrent state-space models (SSMs) and masked attention. Specifically, we formalize and generalize the structured state-space duality between simple recurrent SSMs and masked attention mechanisms. We extend the duality from the scalar-identity case to general diagonal SSMs Section 4.1 and show that these models retain the same training-time complexity lower bounds while supporting richer, multiscale dynamics Section 4.3. We further provide a necessary and sufficient condition under which an SSM corresponds to a 1-semiseparable masked attention mechanism Section 4.4. Finally, we prove a negative result: this duality does not extend to standard softmax attention due to a rank explosion in the induced kernel Section 5. Together, these results strengthen the theoretical bridge between recurrent SSMs and Transformer-style attention, and broaden the design space for expressive yet efficient sequence modeling.

ETHIC STATEMENT

This paper does not involve human subjects, personally identifiable data, or sensitive applications. We do not foresee direct ethical risks. We follow the ICLR Code of Ethics and affirm that all aspects of this research comply with the principles of fairness, transparency, and integrity.

REPRODUCIBILITY STATEMENT

We ensure reproducibility of our theoretical results by including all formal assumptions, definitions, and complete proofs in the appendix. The main text states each theorem clearly and refers to the detailed proofs. No external data or software is required.

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

# A    HIGHER-RANK EQUIVALENCE BETWEEN SEMISEPARABLE MATRICES AND MATRICES WITH SSS REPRESENTATION.

**Proof of Proposition 4.1.**    Here is the main proof of Proposition 4.1.

*Proof.*  Our proof consists of two parts.

**Part 1.**    In this part, we take three steps to show that any lower triangular matrix with an $N$-SSS representation is $N$-semiseparable.

- **Step 1. Express $M$ with its $N$-SSS representation.** Suppose $M \in \mathbb{R}^{T \times T}$ is a lower triangular matrix with an $N$-SSS representation

$$M_{j,i} = c_j^\top A_j \cdots A_{i+1} b_i, \tag{A.1}$$

  for all $i, j \in [T]$, where $b_1, \cdots, b_T, c_1, \cdots, c_T \in \mathbb{R}^N$ and $A^1, \cdots, A^T \in \mathbb{R}^{N \times N}$.

- **Step 2. For any submatrix $S$ whose entries are all on or below the diagonal of $M$, express $S$ with the $N$-SSS representation of $M$.** Suppose $S$ is a submatrix of $M$ such that each entry of $S$ is on or below the principal diagonal line of $M$, we have

$$S = M_{j_1:j_2, i_1:i_2},$$

  for some $1 \le j_1 < j_2 \le T+1$, $1 \le i_1 < i_2 \le T+1$ and $i_2 \le j_1$.
  Let $S^1 \in \mathbb{R}^{(j_2 - j_1) \times N}$ be such that $S^1[:,j] = c_{j+j_1-1}^\top A^{(j+j_1-1)} \cdots A^{j_1+1}$ for $j \in [j_2 - j_1]$. Let
  $S^2 \in \mathbb{R}^{N \times (i_2 - i_1)}$ be such that $S^2[i,:] = A^{j_1} \cdots A^{(i+i_1)} b_{i+i_1-1}$ for $i \in [i_2 - i_1]$.
  Then according to (A.1), we have

$$S = S_1 \cdot S_2.$$

- **Step 3. Upperbound the rank of $S$ with $S^1$ and $S^2$.** Since $S^1$ and $S^2$ both have rank at most $N$, we deduce that $S$ has rank at most $N$.

**Part 2.**    In this part we take $4$ steps to show that any $N$-semiseparable lower triangular matrix has an $N$-SSS representation. Suppose $M \in \mathbb{R}^{T \times T}$ is an $N$-semiseparable lower triangular matrix.

- **Step 1. Divide $M_{t:,:t+1}$ into the product of $2$ matrices of rank at most $N$.** Since $M$ is $N$-semiseparable, $M_{t:,:t+1} \in \mathbb{R}^{(T-t+1) \times t}$ has rank at most $N$ for each $t \in [T]$. Therefore there exist low-rank matrices $W_t \in R^{(T-t+1) \times N}$ and $U_t \in \mathbb{R}^{N \times t}$ such that

$$M_{t:,:t+1} = W_t \cdot U_t. \tag{A.2}$$

We provide a visualization in Figure 3.

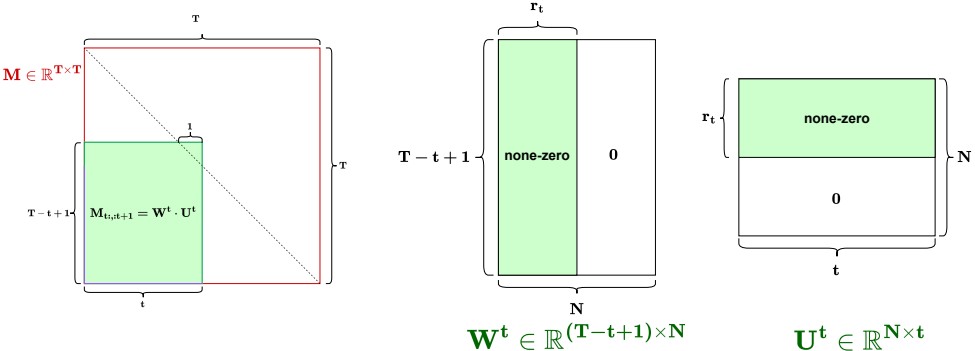

Figure 3: $M_{t:,:t+1} = W^t \cdot U^t$

Figure 4: Only the first $r_t$ columns (rows) of $W^t$ ($U^t$) are non-zero.

Let $r_t \le N$ denote the rank of $M_{t:,:t+1}$ for all $t \in [T]$. Without loss of generality, we construct $W_t$ and $U_t$ to be such that only the first $r_t$ columns of $W_t$ are none-zero, and similarly, only the first $r_t$ rows of $U_t$ are none-zero. We provide a visualization in Figure 4.

This means that the span of $W_t$'s column vectors equals the span of $M_{t:T+1,1:t+1}$ column vectors, and the span of $U^t$'s row vectors equals the span of $M_{t:T+1,1:t+1}$ row vectors.

- **Step 2.** **Suppose $M$ has $N$-SSS representation, analyze the condition $A^1, \cdots, A^T, b_1, \cdots, b_T, c_1, \cdots c_T$ should satisfy.** Note that if $M$ has an $N$-SSS representation, then for all $t \in [T]$, we have

$$
M_{t:,:t+1} = \begin{pmatrix} b_t^\top \\ b_{t+1}^\top A^{t+1} \\ b_{t+2}^\top A^{t+2} A^{t+1} \\ \vdots \\ b_T^\top A^T \cdots A^{t+1} \end{pmatrix} \cdot \begin{pmatrix} A^t \cdots A^2 c_1 & A^t \cdots A^3 c_2 & \cdots & A^t c_{t-1} & c_t \end{pmatrix}. \quad \text{(A.3)}
$$

We expect there exist $A^{1:T+1}$, $b_{1:T+1}$, $c_{1:T+1}$ satisfying

$$
\begin{pmatrix} b_t^\top \\ b_{t+1}^\top A^{t+1} \\ b_{t+2}^\top A^{t+2} A^{t+1} \\ \vdots \\ b_T^\top A^T \cdots A^{t+1} \end{pmatrix} = W^t, \quad \text{(A.4)}
$$

and

$$
\begin{pmatrix} A^t \cdots A^2 c_1 & A^t \cdots A^3 c_2 & \cdots & A^t c_{t-1} & c_t \end{pmatrix} = U^t, \quad \text{(A.5)}
$$

for all $t \in [T]$.

Set $W^{t'} = W_{2:,:}^t$ and $U^{t'} = U_{:,:t}^t$ for all $t \in [T]$, i.e. $W^{t'}$ is $W^t$ without the first row, and $U^{t'}$ is $U^t$ without the last column. We provide a visualization in Figure 5.

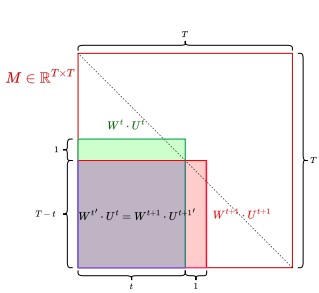

Figure 5: $W^{t'} \cdot U^t = W^{t+1} \cdot U^{t+1'}$

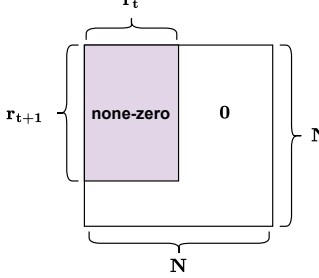

Figure 6: Set only the first $r_t$ columns and the first $r_{t+1}$ rows of $A^{t+1}$ and $A^{t+1'}$ to be non-zero.

Then we have $W^{t+1} \cdot U^{t+1'} = M_{t+1:,:t+1} = W^{t'} \cdot U^t$.

(A.4) and (A.5) requires $W^{t'} = W^{t+1} \cdot A^{t+1}$ and $U^{t+1'} = A^{t+1} \cdot U^t$ for all $t \in [T-1]$.

- **Step 3. Verify the existence of $A^t$ satisfying the conditions mentioned above.** Next we show that there exists $A^{t+1} \in \mathbb{R}^{N \times N}$ for all $t \in [T-1]$ satisfying $W^{t'} = W^{t+1} \cdot A^{t+1}$ and $U^{t+1'} = A^{t+1} \cdot U^t$.

Since the column vectors of $W^{t+1}$ span to be the linear space containing all column vectors of $M_{t+1:,:t+2}$, which also contains all column vectors of $W^{t'}$, there must exist $A^{t+1} \in \mathbb{R}^{N \times N}$ satisfying $W^{t'} = W^{t+1} \cdot A^{t+1}$.

For the same reason there exists $A^{t+1'} \in \mathbb{R}^{N \times N}$ satisfying $U^{t+1'} = A^{t+1'} \cdot U^t$.

Without loss of generality, we set only the first $r_t$ columns and the first $r_{t+1}$ rows of $A^{t+1}$ and $A^{t+1'}$ to be none-zero. We provide a visualization in Figure 6.

Now we deduce that

$$
W^{t+1} \cdot A^{t+1'} \cdot U^t = W^{t+1} \cdot U^{t+1'}
$$

$$= M_{t+1:,:t+1}$$
$$= W^{t'} \cdot U^t$$
$$= W^{t+1} \cdot A^{t+1} \cdot U^t,$$

where the 1st step is by $U^{t+1'} = A^{t+1'} \cdot U^t$, the 2nd and 3rd step is by simple algebra (see Figure 5), and the last step is by $W^{t'} = W^{t+1} \cdot A^{t+1}$. We provide a visualization in Figure 7.

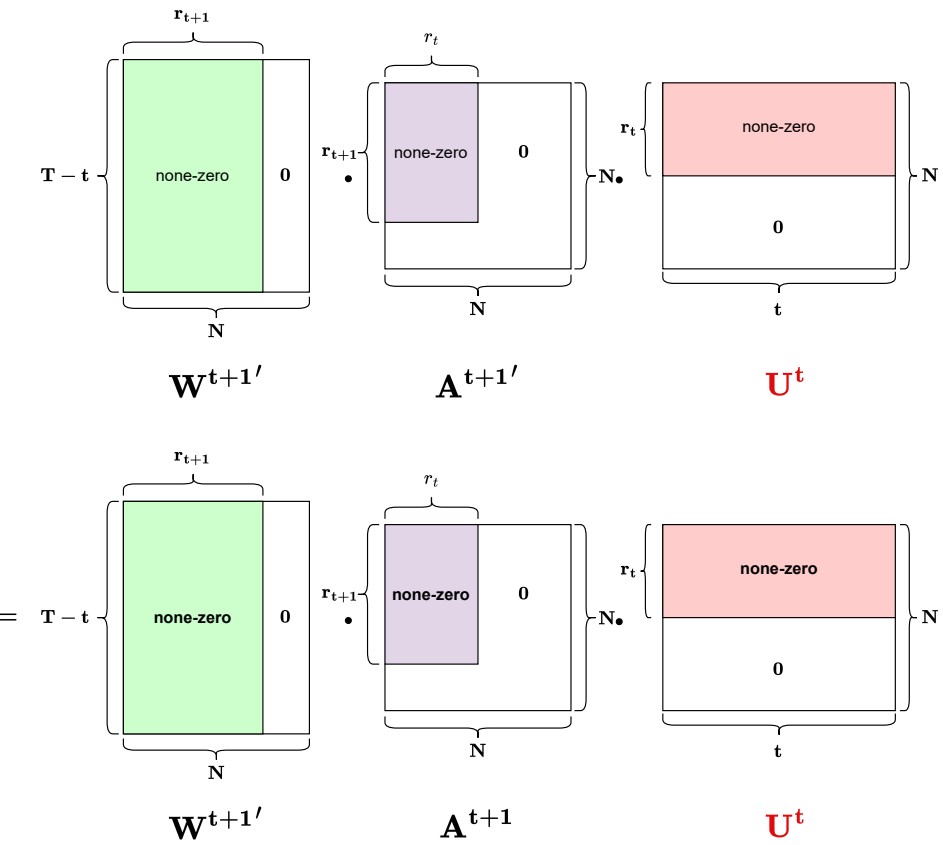

Figure 7: $W^{t+1'} \cdot A^{t+1'} \cdot U^t = W^{t+1'} \cdot A^{t+1} \cdot U^t$

Since $W^{t+1}$ and $U^t$ have rank $r_{t+1}$ and $r_t$ perspectively, we deduce that $A^{t+1'} = A^{t+1}$. Therefore for any $t \in [T-1]$, we have constructed $A^{t+1}$ satisfying both $W^{t'} = W^{t+1} \cdot A^{t+1}$ and $U^{t'} = A^{t+1'} \cdot U^{t+1}$.

- **Step 4. Construct the $N$-SSS representation of $M$ using $A^t$, $W^t$ and $U^t$.** For all $t \in [T]$, let $b_t$ be the first column of $(W^t)^\top$ and $c_t$ be the last column of $U^t$. Let $A^{t+1}$ be as constructed above for $t \in [T-1]$ and $A^1 = I_N$. Then $M_{j,i} = c_j^\top A^j \cdots A^{i+1} b_i$, i.e., $M$ has an $N$-SSS representation.

This completes the proof. □

