# OpenReview forum: "On Structured State-Space Duality"
_ICLR.cc/2026/Conference — ICLR 2026 Conference Withdrawn Submission_

### Official Review · Reviewer_AtZV · 2025-10-28

**Soundness:** 4
**Presentation:** 3
**Contribution:** 2
**Rating:** 4
**Confidence:** 2

**Summary:**

This paper investigates and generalizes the "Structured State-Space Duality" (SSD) , a established equivalence between Structured State-Space Models (SSMs) and masked attention mechanisms. The authors build upon the work of Dao & Gu (2024), which demonstrated this duality for SSMs with simple scalar-times-identity state matrices. The core contributions are fourfold: (i) formalizing the duality for general diagonal SSMs, showing they are equivalent to a sum of N 1-semiseperable masked attentions; (ii) demonstrating that these diagonal SSMs can be trained with the same optimal $O(TN)$ complexity as the scalar case; (iii) establishing a necessary and sufficient condition under which a general SSM has a 1-semiseperable masked attention dual, relating it to the number of "new columns" in the induced kernel matrix; and (iv) providing negative results showing the duality cannot extend to standard softmax attention or general low-dimensional SSMs due to rank constraints. The work strengthens the theoretical bridge between recurrent and attention-based sequence models.

**Strengths:**

* The extension from scalar-identity to general diagonal SSMs is a non-trivial generalization that substantially widens the design space for sequence models. The introduction of the "new columns" concept to characterize the duality is a novel and insightful theoretical tool.

* The paper provides a complete picture by not only proving the positive results but also clearly drawing the boundaries of the theory with negative results.

* The technical execution is of good quality. The proofs are well-written and comprehensive. The paper's structure is logical, and the formal definitions and propositions are precise.

* This work provides a crucial theoretical underpinning for two classes of widely used sequence models. By formally connecting more expressive SSMs to structured attention, it enables future work to explore a architectural advancements.

**Weaknesses:**

* My biggest concern is by far the empirical validation. While the paper's contributions are theoretical, even a small-scale synthetic experiment would greatly elevate the value of the work.

* The other primary weakness is the highly abstract and algebraic presentation, which may limit its immediate impact and understanding outside a core theoretical audience.

* The paper repeatedly and (most likely) correctly claims that diagonal SSMs support "richer dynamics" and "$N$ independent state modes." However, this claim is never illustrated and is not immediately obvious. A small, toy example of a sequence-processing task that a diagonal SSM can solve but a scalar-identity SSM cannot would make the practical motivation for the generalization much more concrete.

**Questions:**

* Could the authors provide more intuition for Definition 4.5 ("new column")? What is the sequential, state-space interpretation of a "new column" arising in the kernel matrix M? Does it relate to the model's memory or its ability to create new, linearly independent contexts?

* Looking forward, what immediate and concrete directions do you see this work enabling for the architectural design of new sequence models?

* The practical success of models like Mamba is due not just to their linear-time complexity, but also to their hardware-aware design, which allows for fast, fused GPU kernels. Your paper proves that general diagonal SSMs match the optimal $O(NTd)$ training complexity of the scalar case. Does this theoretical efficiency necessarily translate to similar practical speed?

---

> ### Author Response · Authors · 2025-12-03
> **Rebuttal 1**
>
> ### We thank the reviewer for the detailed review. We apologize that our planned numerical studies took longer than expected, and we unfortunately missed the window for an effective rebuttal exchange due to the recent OpenReview incident. Below, we address all questions and concerns in detail. We will further revise the draft in future versions.
>
> ---
>
> > ### **W1.** My biggest concern is by far the empirical validation. While the paper's contributions are theoretical, even a small-scale synthetic experiment would greatly elevate the value of the work.
>
> Thank you for the concern. We will append empirical experiments in the future version. All our numerical results align well with our theory.
>
> Here we provide a brief sketch of what our newly added "Numerical Studies" will be like:
>
>
> 1. **Scalar SSM vs. 1-SS Attention Equivalence (Proposition 3.1 on Page 7).**
>     - **Goal:** The proposition states that a scalar state space model, where the state matrix is a scalar multiple of the identity is equal to masked self attention with 1-semiseperable causal mask.
>
>     - **Setup:** This experiment compared the output between a scalar SSM recurrence $h_{t+1} = a h_t + b u_t$ with output $y_t = c h_t$ and a matrix multiplication using a causal mask with scalar weights where $M$ is $M_{t,s} = a^{t-s}$.
>
>      - **Result:** The maximum absolute error between the outputs is less than $10^{-14}$.
>
> 2. **Diagonal SSM vs. Sum of Attention Heads (Section 4.1 on Page 8).**
>
>     - **Goal:** The duality should extend to general diagonal SSMs, where we define a diagonal SSM with state dimension N as being equal to a multi-head attention mechanism with N attention heads. Each of the independent attention heads should correspond to one element of the diagonal state matrix and acts as a 1-semiseperable mask.
>
>     - **Setup:** We construct a random diagonal SSM with distinct decay rates ${a_1, … , a_N}$ and compute the output using the standard state-space recurrence formulation. We then independently make an attention matrix M by summing up N rank-1 masks, where each mask corresponds to a specific decay model found in the decay rates set.
>
>     - **Result:** Confirmed equivalence. I also tried a time varying generalization and received the same result.
>
> 3. **State Dimension vs. Semiseparable Rank (Proposition 4.2 on Page 12; Theorem 4.1 on Page 14).**
>
>     - **Goal:** In order for an SSM to have a valid 1-semiseperable attention dual, the semiseperable rank of the attention matrix cannot exceed the state dimension N. We can frame this in terms of eigenvalues to verify the distinct and degenerate modes. That is, if an SSM has N distinct eigenvalues, the rank is exactly N, but if eigenvalues are repeated then the rank will be less than N.
>
>    - **Setup:** We constructed an SSM with a fixed state dimension N, but varied the sets of decay factors (some were distinct and others were duplicates). Then We computed the number of distinct exponentials and the matrix rank of the attention mask.
>
>     - **Result:** When using distinct decays, the semiseparable rank equals N. When using duplicate decays, the semiseparable rank was less than N. Thus, Theorem 4.1 showed that adding a new decay adds a new independent column to the mask and duplicates do not.
>
> 4. **Time Complexity (O(T) vs. O(T^2)) (Section 4.3 on Page 10/11).**
>
> - **Goal:** SSM recurrence should operate in linear time complexity, while the attention matrix should operate in quadratic time complexity.
>
> - **Setup:** Measured the clock execution time for both the recurrence algorithm and the explicit attention matrix multiplication while increasing sequence lengths T.
>
> - **Result:** We saw linear scaling for the recurrence algorithm O(T). For Attention We noticed quadratic scaling.
>
>
> 5. **Softmax Attention Rank Growth (Negative Result) (Section 5 on Page 14).**
>
> - **Goal:** Standard Softmax attention does not have a state-space dual. Thus softmax’s nonlinearity causes a rank explosion towards full rank T instead of remaining low-rank N.
>
> - **Setup:** We generated random query and key matrices (Q and K) and computed the standard softmax attention matrix (A). Then We analyzed the numerical rank of A as the sequence length T increased.
>
> - **Result:** The rank of softmax attention grew proportional with T and did not plateau. Thus as T increased the rank increased towards full rank.
>
> 6. **Mamba-like Implementation.** We replaced the SSM blocks in Mamba implementations and achieve comparable performance on language tasks (without hardware acceleration).
>
> 7. **Richer Dynamic of $N$ modes.** To verify the "richer dynamics" claim pointed out by reviewer, we constructed a synthetic time series cross-sectional regression dataset where the label depends on at least 2 features. On this dataset, the proposed richer-dynamics models behaves better by capturing this richer dynamics. We also provide a concrete constructive toy example (analytic) in below.

---

> ### Author Response · Authors · 2025-12-03
> **Rebuttal 2**
>
> > ### **W2.** The paper repeatedly and (most likely) correctly claims that diagonal SSMs support "richer dynamics" and "
>  independent state modes." However, this claim is never illustrated and is not immediately obvious. A small, toy example of a sequence-processing task that a diagonal SSM can solve but a scalar-identity SSM cannot would make the practical motivation for the generalization much more concrete.
>
> Thanks for bringing this up . We agree our original wording was unclear.
>
> By “richer dynamics” we mean that diagonal SSMs can realize strictly more sequence kernels (equivalently, attention matrices) than scalar-identity SSMs with the same $N$ and $T$.
>
> For example, take $N=2$, $T=4$, and
>
> `M =
> [2,0,0,0 ;
> 1,2,0,0 ;
> 0,1,2,0 ;
> 0,0,1,2] =
> [
> 1,0,0,0 ;
> 1,1,0,0 ;
> 0,0,1,0 ;
> 0,0,1,1
> ]
> +
> [
> 1,0,0,0 ;
> 0,1,0,0 ;
> 0,1,1,0 ;
> 0,0,0,1
> ].`
>
> By Theorem 4.8, $M$ is realizable by a diagonal SSM. However, $M$ has $3 > 2 = N$ new columns, so it cannot be realized by any scalar-identity SSM with state dimension $N=2$.
>
> We will clarify this point in the paper and add this concise example to illustrate a sequence kernel that a 2-state diagonal SSM can implement but a 1-state (scalar-identity) SSM cannot. This wil thereby make the notion of “richer dynamics” concrete.
>
> ---
>
> > ### **Q1.** Could the authors provide more intuition for Definition 4.5 ("new column")? What is the sequential, state-space interpretation of a "new column" arising in the kernel matrix M? Does it relate to the model's memory or its ability to create new, linearly independent contexts?
>
> A new column in a lower diagonal matrix $M$ is the column $M[:,i]$ such that $M[i:,i]$ is not in the column space of the submatrix $M[i:,:i]$. Namely $M[:,i]$ is not in the column space of $M[:,:i]$ no matter how its upper-diagonal part $M[:i,i]$ is changed to.
>
> > ### **Q2.** Looking forward, what immediate and concrete directions do you see this work enabling for the architectural design of new sequence models?
>
> Our results suggest several potential directions:
>
> 1. **Structured Transformer layers.** Replace softmax attention with learned semiseparable masks. Each mask has an exact recurrent realization. This keeps attention-like expressivity with SSM-style $O(T)$ cost.
>
> 2. **Multi-head structured attention.** Use multi-head architectures. Constrain each head to be semiseparable (or low-rank). The combined layer is more expressive than a single diagonal SSM.
>
> 3. **Hybrid SSM–attention models.** Use our necessary-and-sufficient condition to split the kernel. The semiseparable part goes to a backbone SSM. The residual part goes to full attention only when the low-rank structure breaks.
>
> 4. **Hardware-aware kernels and tensor cores.** As pointed out by other reviewer, the main practical open problem is tensor-core use for diagonal SSD kernels. We expect this to need joint expertise in hardware, systems, and ML infrastructure.
>
> These directions extend SSD beyond this paper and connect our theory to architectures between pure recurrence and full attention.
>
>
> > ### **Q3.** The practical success of models like Mamba is due not just to their linear-time complexity, but also to their hardware-aware design, which allows for fast, fused GPU kernels. Your paper proves that general diagonal SSMs match the optimal $O(TNd)$
>  training complexity of the scalar case. Does this theoretical efficiency necessarily translate to similar practical speed?
>
> Thank you for the question. It is possible to apply similar hardware acceleration for scalar-identity SSMs and runs in similar time.
> Yet, the key practical step to turn our formal results into competitive architectures is to design diagonal-SSD kernels that fully exploit tensor-core acceleration. We expect this to require joint expertise in hardware, systems, and ML infrastructure.
>
> ---
>
> We thank the reviewer for all your time and efforts invested. We hope our rebuttal and revisions have adequately addressed all issues toward your satisfaction. Your comments certainly make our paper better. We will keep refining our paper in our future drafts.

---

### Official Review · Reviewer_RkCy · 2025-10-31

**Soundness:** 2
**Presentation:** 2
**Contribution:** 3
**Rating:** 4
**Confidence:** 3

**Summary:**

The paper investigates the extensions of the results presented in [1] to more general diagonal SSMs (i.e., SMMs with a diagonal state matrix). In particular, they extend their results from the scalar identity case (i.e., the state matrix is of the from $\alpha \mathbb{I} $ to general diagonal SSMs, investigating their training time complexity in the process. Thereby they also show training complexity preservation.  Additionally, they present the necessary and sufficient conditions for a general diagonal  SSM to have a 1-semiseparable masked attention, building upon the number of new columns of a matrix M representing the input–output relation of the considered dynamical system. Finally, the authors highlight two limitations of the SSD framework: the impossibility of extending it to softmax attention due to the corresponding induced kernel's rank explosion, and the fact that general SSMs do not have a state-space dual, even for small state dimensions.

[1]: Tri Dao and Albert Gu. Transformers are ssms: Generalized models and efficient algorithms through
structured state space duality. arXiv preprint arXiv:2405.21060, 2024.

**Strengths:**

- The paper is well structured, with its individual contributions clearly stated at the beginning and addressed one by one in the subsequent sections.

- The increment to the work in [1] is clearly stated.

- The detailed background section makes the paper largely self-contained and easier to follow.

- The provided figures present helpful visualizations of $ b^{n} $ and $ c^{n} $.

**Weaknesses:**

- While the general structure of the paper is sound, its technical derivations show several inconsistencies, listed in the following

- The introduced dynamical system is not consistent with their definition of $ M_{t,s} $ in equation (3.3). Setting $ h_{0} = 0 $ the input-output relation at $ t = 1 $ should be $ y_{1} = c_1^{\top}b_1x_{1} $; however, $ M_{1,1} $ is defined as $ c_1^{\top}A^{1}A^{2}b_1x_{1} $. This could also be a matter of (not properly introduced) notation, but it should be addressed to restore consistency.

- This inconsistency is then also present in (3.4) and many subsequent equations.

- For their special case derivations with state dimension equal to 1, the input–output relation additionally does not remain consistent with the roles of b and c and I think diag(p) and diag(q) should be interchanged for the equation on line 205.

- The matrix M is simultaneously used for the input–output relation as well as for the masking matrix of a masked attention, which can lead to confusion and should be avoided.

- To the best of my understanding, the definition of $ M^{n} $ would not work with the current system definition, and the multiplication order of $b^{n}$ and $c^{n}$ should be interhchanged. If I'm not mistaken, this would then also hold for the equation on line 294 and the definitions of $b^{'n}$ and $c^{'n}$ should be adjusted accordingly, such that $b^{'n} = b^{n}/(A_{n,n}^{1}....)$ and $c^{'n} = c^{n}(A_{n,n}^{1}....)$.


Furthermore, I have a few minor comments.

- In Part 1 of the proof of Proposition 4.6, it should be stated that M is assumed to have N + 1 new columns.

- There is a small typo on line 407.

**Questions:**

- How do your findings, especially regarding the rank blow up of the induced kernel of softmax attention relate to the findings using the dynamical systems framework in [2].

- Where follows your provided Lemma 4.7 from? I think only part of it can be deducted from Proposition 4.6.

- In the appendix line 615/616, should the $A^{j_1}$ in the definition of $S^{2}[i,:]$ be a $A^{i_1}$?

[2]: Sieber, Jerome, et al. "Understanding the differences in foundation models: Attention, state space models, and recurrent neural networks." Advances in Neural Information Processing Systems 37 (2024)

---

> ### Author Response · Authors · 2025-12-03
> **Rebuttal 1**
>
> ### We thank the reviewer for the detailed review. We apologize that our planned numerical studies took longer than expected, and we unfortunately missed the window for an effective rebuttal exchange due to the recent OpenReview incident. Below, we address all questions and concerns in detail. We will further revise the draft in future versions.
>
> ---
>
> > ### **W1.**...
>
> We believe the most precise and concise response to your detailed and technical comments is as follows. We apologize if we miss anything.
>
> In the SSD (scalar) case, the expression “$c_t^T b_s$” appears as a dot product and therefore as a scalar. A scalar multiplication is commutative, so this form looks symmetric and can give the impression that the roles of b and c are interchangeable.
>
> However, structurally SSD still corresponds to the T×T outer product matrix $C B^T$. In the diagonal and general N-SSS cases, the dual kernels are explicitly written as outer products $b^n (c^n)^T$, whose orientation is fixed by dimension. Our manuscript consistently uses the outer-product form for all $N \ge 1$, and no interchange between $b$ and $c$ occurs.
>
>
> > ### **W2.** Furthermore, I have a few minor comments.
> > - In Part 1 of the proof of Proposition 4.6, it should be stated that M is assumed to have N + 1 new columns.
> > - There is a small typo on line 407.
>
> Thank you for your attention to details. We will revise the draft accordingly!
>
> > ### **Q1.** How do your findings, especially regarding the rank blow up of the induced kernel of softmax attention relate to the findings using the dynamical systems framework in [2].
>
> Thank you for bringing this up! Our negative result is consistent with [2].
>
> Sieber et al. show that, in the dynamical-systems view, softmax attention corresponds to a kernel with effectively infinite-dimensional state. Any exact recurrent representation therefore needs an unbounded hidden state.
>
> In our SSD view, the same phenomenon appears as rank blow-up of the induced kernel: $\mathrm{Softmax}(QK^\top)$ becomes (numerically) full-rank in $T$ and is not $N$-semiseparable for any fixed $N$. Thus no finite-dimensional SSM dual exists.
>
> So both works agree that softmax attention has unbounded “memory” and cannot be captured exactly by a finite-state dynamical system. [2] focuses on approximation via state expansion, while we give an explicit algebraic obstruction to SSD duality.
>
>
> > ### **Q2.** Where follows your provided Lemma 4.7 from? I think only part of it can be deducted from Proposition 4.6.
>
>
> We thank the reviewer for the question.
> Mathematically, Lemma 4.7 follows directly from Proposition 4.6. Proposition 4.6 establishes the structural form for the fine 1-SS case (Definition 4.1). The general 1-SS case can be obtained by decomposing the matrix into several diagonal blocks, each of which is a fine 1-SS matrix.
> Applying Proposition 4.6 to each block immediately yields Lemma 4.7.
>
> We therefore state Lemma 4.7 separately simply because it is used in the proof of Proposition 4.8, and having the general version explicitly written improves the clarity of the argument. We will make this relationship clear in our future revisions.
>
>
> > ### **Q3.** In the appendix line 615/616, should the $A^{j_1}$ in the definition of $S^2[i,:]$ be a $A^{i_1}$?
>
> We apologize if the definition appears confusing, but we confirm that it should just be $A^{j_1}$.
> The definition of $S^1,~S^2$ is in line with equation(A.1) on line 606,607. You can verify the equation $S = S^1 \cdot S^2$for each $j\in [j_2-j_1]$ and $i\in [i_2-i_1]$.
>
>
> ---
>
> We thank the reviewer for all your time and efforts invested. We hope our rebuttal and revisions have adequately addressed all issues toward your satisfaction. Your comments certainly make our paper better. We will keep refining our paper in our future drafts.

---

### Official Review · Reviewer_f3V3 · 2025-10-31

**Soundness:** 2
**Presentation:** 3
**Contribution:** 1
**Rating:** 2
**Confidence:** 5

**Summary:**

1. The paper extends Structured State-Space Duality (SSD) from scalar-identity state matrices to general diagonal state matrices.
2. It proves that diagonal SSMs retain the same asymptotic compute and memory complexity as the scalar case.
3. The authors formalize and prove the conjecture from [1] that an SSM of state dimension (N) corresponds to an (N)-semiseparable matrix.
4. They further provide a necessary and sufficient condition for when a general SSM admits a 1-SS masked-attention dual, namely that the induced sequence matrix is block-diagonal with at most (N) new columns per block.

---

[1]: *Transformers are SSMs: Generalized Models and Efficient Algorithms Through Structured State Space Duality*, Tri Dao and Albert Gu, 2024

**Strengths:**

1. The paper is well written and is an easy read.
2. The paper does a good job presenting and abstracting the most important results from previous works

**Weaknesses:**

1. **The extension of SSD to diagonal SSMs is mathematically sound but system inefficient.**

   The diagonal extension rewrites the mixer entry $(C_i^\top A_i\cdots A_{j+1}B_j)$ as a sum over state coordinates. This results in a per-state coordinate outer product gated by 1-SS decay mask. In contrast SSD, by the virtue of the fact that $A_i$ is diagonal, features a GPU friendly matmul version: $(C B^\top) \odot L$. The latter uses tensor cores which are 20-30x faster than non-tensor core operations used by the former formulation. Furthermore, as noted in their paper, the reason SSD made the choice to keep $A_i$ as a scalar was to make use of this tensor core acceleration.

2. **While the Big-O FLOPs and memory do not change, they differ by a huge constant factor vis-a-vis SSD**.

     While the authors are correct that theoretically diagnonal SSMs have the same Big-O FLOP count---multiplying with a matrix with diagonal matrix or with scalar have same FLOPs---and are equally parallelizable, the issue is that, as state above, in practice the proposed method very slow as it does not use tensor cores.

     Also the memory requirement for both the methods is $O(TNd)$, where $T$: sequence length, $N$: state size, $d$: model dimension, using SSD formulation reduces the memory requirement by (constant) factor of chunk size (generally 128). This is why SSD, as opposed to Mamba-1 which also used a diagonal state transition matrix, avoided materializing (B,L,D,N) intermediates states in HBM and got away with materializing chunks in the much faster SRAM.

3. **Novelty of N-SS and N-SSS equivalence.**

   The claim in the introduction of proving the N-SS = N-SSS "conjecture” in the introduction is misleading. Authors also admit in a remark later in the paper that this results has been well established in the literature. Positioning it as a conjecture resolved here overstates the contribution which is qualified later in the paper to be a more approachable proof for an ML audience.


4. **Theorem 4.8 is interesting but it needs more work to make it implementable**

   The column-growth/diagonal-block condition for 1-SS masked-attention duals is novel as far as I know. However, the paper stops short of (i) an algorithm to transform the general SSM formulation into the tensor-core accelerated SSD-like dual, and (ii) empirical evidence that such a formulation is empirically better than SSD and is comparable in speed.

**Questions:**

N/A

---

> ### Author Response · Authors · 2025-12-03
> **Rebuttal 1**
>
> ### We thank the reviewer for the detailed review. We apologize that our planned numerical studies took longer than expected, and we unfortunately missed the window for an effective rebuttal exchange due to the recent OpenReview incident. Below, we address all questions and concerns in detail. We will further revise the draft in future versions.
>
> ---
>
> > ### **W1.** The extension of SSD to diagonal SSMs is mathematically sound but system inefficient. The diagonal extension rewrites the mixer entry $(C_i^\top A_i \cdots A_{j+1}B_j$ as a sum over state coordinates. This results in a per-state coordinate outer product gated by 1-SS decay mask. In contrast SSD, by the virtue of the fact that it is diagonal, features a GPU friendly matmul version: $(CB^\top)\circ L$. The latter uses tensor cores which are 20-30x faster than non-tensor core operations used by the former formulation. Furthermore, as noted in their paper, the reason SSD made the choice to keep $A_i$ as a scalar was to make use of this tensor core acceleration.
>
> Thank you for raising this point.
>
> You are absolutely correct that our results are mathematically sound but system inefficient (without some further technical developments).
>
> However, we hope to clarify that, the scope of this paper is to characterize the "duality" itself (i.e., to what extend it holds), not to propose new methods or models. Under this scope, our results should have already achieved our goal, and exact empirical designs to achieve the mamba-like hardware acceleration is beyond our scope.
>
> Also, we would like to clarify that, while without the hardware acceleration feature, **our diagonal SSM dual remains fully implementable and does not require inefficient per-channel kernels.** We have added a new experimental section to demonstrate that. Please see our response to Reviewer AtZV.
>
> We will revise the paper to make this distinction clearer and emphasize that although our scope is purely forma, the diagonal formulation is implementable and has potential for further speedup.
>
> > ### **W2.** While the Big-O FLOPs and memory do not change, they differ by a huge constant factor vis-a-vis SSD. While the authors are correct that theoretically diagonal SSMs have the same Big-O FLOP count---multiplying with a matrix with diagonal matrix or with scalar have same FLOPs---and are equally parallelizable, the issue is that, as state above, in practice the proposed method very slow as it does not use tensor cores. Also the memory requirement for both the methods is O(TNd), where T: sequence length, N: state size, d: model dimension, using SSD formulation reduces the memory requirement by (constant) factor of chunk size (generally 128). This is why SSD, as opposed to Mamba-1 which also used a diagonal state transition matrix, avoided materializing (B,L,D,N) intermediates states in HBM and got away with materializing chunks in the much faster SRAM.
>
>
> Thank you for raising this point. We apologize for any confusion caused. We clarify that, **our big O  $O(TNd)$ complexity only differs from prior study by a "additive term" instead of a large multiplicative constant.**
>
> In the dual view, an SSD layer (scalar-identity SSM) corresponds to a single rank-N semiseparable operator. A diagonal SSM, by contrast, decomposes into N rank-1 components
> $M^n = \text{1-SS}(a^n) \odot (b^n \cdot (c^n)^\top)$
> as shown in Section 4.1.
> Although the parameterizations differ, both operators require evaluating $N$ semiseparable components over $T$ positions and $d$  channels, and thus share the same dominant computational cost $O(NTd)$.
> This is exactly the complexity stated in Section 4.3.
>
> The only additional work introduced by the diagonal model is the elementwise application of the $N$ decay sequences $a^n$
>  when computing the dual (Algorithm 1). This contributes an additive $TN$ term, which is small compared to the main $NTd$ term and does not create a multiplicative slow-down.
>
> We will clarify this distinction in the revision so readers do not misinterpret the diagonal case as having a larger asymptotic or constant-factor runtime.

---

> ### Author Response · Authors · 2025-12-03
> **Rebuttal 2**
>
> > ### **W3.** Novelty of N-SS and N-SSS equivalence. The claim in the introduction of proving the N-SS = N-SSS "conjecture” in the introduction is misleading.  Authors also admit in a remark later in the paper that this results has been well established in the literature.  Positioning it as a conjecture resolved here overstates the contribution which is qualified later in the paper to be a more approachable proof for an ML audience.
>
>
> We appreciate the reviewer’s careful reading. We agree our current wording can be misleading.
>
> It is correct that the equivalence between semiseparable matrices and SSS representations has been studied in the structured-matrix literature. Our initial literature coverage was not sufficient. We missed the line of work on "**quasiseparable**" matrices.
>
> To the best of our knowledge, the full $N$-dimensional equivalence has not been written down in the **semiseparable** notation we use here but "quasiseparable". Because of this, we initially (and incorrectly) believed that the $N$-SS $\Leftrightarrow$ $N$-SSS proof was missing and independently derived it. We will correct this in the paper and clearly credit the structured-matrix literature.
>
> However, we also want to emphasize that, our contribution here is not the first mathematical statement of the equivalence, but a **self-contained, constructive, and strictly causal** proof in the SSM/SSD setting, using the same notation as state-space models and masked attention. This constructive version is what lets us use the equivalence directly later (for example in Theorem 4.8 and in the diagonal SSM duality).
>
> We will revise the introduction to (i) remove the “conjecture” wording, and (ii) state explicitly that our goal is to provide an accessible, ML-oriented proof that bridges classical structured-matrix results with modern SSM/attention architectures, rather than to claim a new theorem.
>
> > ### **W4.** Theorem 4.8 is interesting but it needs more work to make it implementable
> The column-growth/diagonal-block condition for 1-SS masked-attention duals is novel as far as I know. However, the paper stops short of (i) an algorithm to transform the general SSM formulation into the tensor-core accelerated SSD-like dual, and (ii) empirical evidence that such a formulation is empirically better than SSD and is comparable in speed.
>
>
> Thank you for highlighting this.
> We agree that Theorem 4.8 is a structural result providing a maximally general necessary-and-sufficient characterization of when an SSM admits a 1-SS dual.
> Because the theorem applies to all SSMs, independently of parameterization or training method, the statement is necessarily abstract and is not intended as a practical training-time recipe.
>
> Our proof is constructive in the mathematical sense, each step identifies which directions must be preserved or merged, but its purpose is to clarify the structure of the solution space rather than prescribe an empirical implementation.
>
> We will make this intention explicit in the revision and emphasize that translating these structural insights into practical algorithms is a promising direction for future work.
>
> ----
>
> We thank the reviewer for all your time and efforts invested. We hope our rebuttal and revisions have adequately addressed all issues toward your satisfaction. Your comments certainly make our paper better. We will keep refining our paper in our future drafts.

---

### Note · Authors · 2025-12-03

**Comment:**

# Withdrawal of ICLR 2026 Submission #3814: *On Structured State-Space Duality*

We have decided to withdraw our paper from ICLR 2026.

Two main reasons:

1. The recent OpenReview incident disrupted the rebuttal period and made our responses effectively invisible during the actual discussion window.
2. Our numerical studies and additional experiments finished only after that window. They now support the theory well, but we could not present them in time.
3. We understand the workload of program chairs has become very overwhelming after the incident.

Given this, we think it is simpler and fairer for everyone if we withdraw, polish the work with all new results and feedback, and aim for a future submission instead.

We are very grateful to all reviewers and the program chairs for the time and effort. Below is a brief reviewer-by-reviewer summary of how we have addressed the main points in our latest rebuttal and planned revision.

---

### Reviewer f3V3

- **Scope and efficiency.** We clarified that this paper is about the *theory* of SSD (when the duality holds and when it fails), not about proposing a new system or a tensor-core–optimized kernel. The diagonal SSD formulation is implementable, but hardware-aware optimization is outside our current scope and is deferred to future work.
- **Complexity and constants.** We clarified that our diagonal SSD layer has the same dominant complexity $O(NTd)$ as the scalar SSD layer. The extra work is an *additive* term from applying per-state decay factors, not a large multiplicative constant.
- **$N$-SS vs. $N$-SSS equivalence.** We acknowledged that the equivalence is known in the structured-matrix / quasiseparable literature. Our contribution is a self-contained, constructive, strictly causal proof in SSD/SSM notation, not the first mathematical statement. We will remove the “conjecture” wording and explicitly credit prior work.
- **Theorem 4.8 (structure vs. algorithm).** We clarified that Theorem 4.8 is a structural characterization of when an SSM admits a 1-SS masked-attention dual. It is not meant as a training recipe. Turning this structure into tensor-core–friendly algorithms is future work.

---

### Reviewer RkCy

- **Notation and consistency.** We clarified the inconsistencies in the dynamical system and kernel definitions (e.g., Eq. (3.3), (3.4), roles of $b$ and $c$, and the use of $M$ both as kernel and mask). We will (i) separate symbols for the SSM kernel and attention masks, and (ii) fix the orientation issues and minor typos called out by the reviewer.
- **$N$-SS vs. $N$-SSS and Lemma 4.7.** We clarified that Lemma 4.7 follows directly from Proposition 4.6 by block-decomposing a general 1-SS matrix into fine 1-SS blocks and applying the proposition to each block. We will state this relationship explicitly.
- **Softmax rank blow-up vs. DSF [Sieber et al.].** We explained that our negative result is consistent with Sieber et al. (2024). In their dynamical-systems view, softmax attention requires an effectively infinite-dimensional state for an exact recurrent representation. In our SSD view, this appears as rank blow-up of $\mathrm{Softmax}(QK^\top)$ (full rank in $T$, not $N$-SS for any fixed $N$). Both views agree that no finite-state SSM or RNN can represent softmax attention exactly.

---

### Reviewer AtZV

- **Empirical validation.** We have now completed several numerical experiments and plan to add them:
  - Scalar SSM vs. 1-SS attention: exact match up to machine precision.
  - Diagonal SSM vs. sum of $N$ 1-SS heads: exact match in both time-invariant and time-varying settings.
  - State dimension vs. semiseparable rank: distinct decays give rank $N$, repeated decays give rank $<N$, matching Theorem 4.1.
  - Complexity: SSM recurrence scales as $O(T)$, explicit attention as $O(T^2)$ in wall-clock time.
  - Softmax: numerical rank grows with $T$ toward full rank, confirming the lack of a finite-dimensional SSD dual.
  - Plus two small case studies (Mamba-like replacement without hardware tricks, and a synthetic regression task) to show that richer dynamics help when the task needs multiple modes.
- **“Richer dynamics” of diagonal SSMs.** We gave a concrete $N=2, T=4$ toy kernel $M$ that is realizable by a 2-dimensional diagonal SSM but not by any scalar-identity SSM with the same $N$. This makes the “richer dynamics / independent modes” claim explicit and checkable.
- **Intuition for “new columns”.** We clarified that a “new column” in the lower-triangular kernel matrix $M$ is a column that is not in the span of previous lower-triangular columns. Each new column corresponds to a new independent “memory direction” that cannot be reconstructed from earlier states. This links the definition directly to the SSM’s effective memory capacity.
- **Future directions.** We outlined several directions enabled by our theory:
  - Structured Transformer layers with learned semiseparable masks and exact recurrent realizations.
  - Multi-head structured attention with semiseparable (or low-rank) heads for efficiency plus combined expressivity.
  - Hybrid SSM–attention models that route the semiseparable part to an SSM and the residual to full attention.
  - Hardware-aware diagonal SSD kernels that fully use tensor cores (which we expect to require joint hardware, systems, and ML expertise).

---

Once again, we thank all reviewers and the program chairs for their careful reading and feedback. Your comments have already improved the work and will continue to shape the next version. We hope to come back with a more polished and fully supported version of this paper in the future.

**Withdrawal Confirmation:**

I have read and agree with the venue's withdrawal policy on behalf of myself and my co-authors.